# PROBING INTO OVERFITTING FOR VIDEO RECOGNITION

## ABSTRACT

Video recognition methods based on 2D networks have thrived in recent years, leveraging the advanced image classification techniques. However, overfitting is a severe problem in 2D video recognition models as 1) the scale of video datasets is relatively small compared to image recognition datasets like ImageNet; 2) current pipeline treats informative and non-informative (e.g., background) frames equally during optimization which aggravates overfitting. Based on these challenges, we design a video-specific data augmentation approach, named *Ghost Motion* (GM), to alleviate overfitting. Specifically, GM shifts channels along the temporal dimension to enable semantic motion information diffused into other frames which may be irrelevant originally, leading to motion artifacts which describe the appearance change and emphasize the motion salient frames. In this manner, GM can prevent the model from overfitting non-informative frames and results in better generalization ability. Comprehensive empirical validation on various architectures and datasets shows that GM can improve the generalization of existing methods and is compatible to existing image-level data augmentation approaches to further boost the performance.

## 1 INTRODUCTION

Video recognition methods has evolved rapidly due to the increasing number of online videos and success of advanced deep neural networks. Even if 3D networks (Feichtenhofer et al., 2019; Carreira & Zisserman, 2017; Tran et al., 2015) provide straightforward solutions for video recognition, 2D based methods (Wang et al., 2016; Lin et al., 2019; Li et al., 2020; Wang et al., 2021) still arouse researchers' interests because of their efficiency in both computing and storage. However, 2D networks still suffer from overfitting issue. For instance, on Something-Something V1 (Goyal et al., 2017) dataset which contains strong temporal dependency, the training and validation accuracy of TSM (Lin et al., 2019) is 81.22% and 45.34%, respectively. Besides, its Expected Calibration Error (ECE) (Guo et al., 2017) is 25.83% which means the model gives overconfident predictions and brings negative impact when deploying the model in real scenarios.

There are two main reasons for overfitting in video: 1) video recognition benchmarks usually have fewer training samples compared with image classification datasets (e.g., ImageNet (Deng et al., 2009) with 1.2 million training samples compared to Kinetics (Kay et al., 2017) with 240K videos). Furthermore, spatial-temporal modeling for video is harder than recognizing static images, which should require even more samples. 2) 2D based methods average the logits of all frames to vote for final decision which increases the tendency to overfit frames which contain less semantic information (e.g., background scene). In this view, these frames can be regarded as noise in optimization because they do not provide motion information for temporal modeling.

To alleviate overfitting, many attempts have been proposed. Dropout (Srivastava et al., 2014) and Label Smoothing (Szegedy et al., 2016) are widely used in deep networks because of the regularization effects they bring. Designing data augmentation methods (Zhang et al., 2017; Yun et al., 2019; DeVries & Taylor, 2017; Cubuk et al., 2020) to relieve overfitting is another line of research. Although some methods have shown effectiveness in image-level tasks, directly employing them on video tasks may result in detrimental temporal representations as these methods are not specifically designed for video data and some transformations will break the motion pattern.

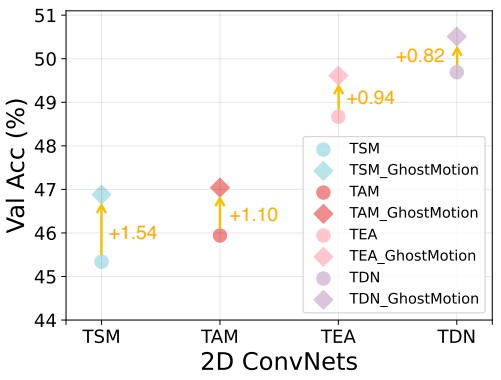 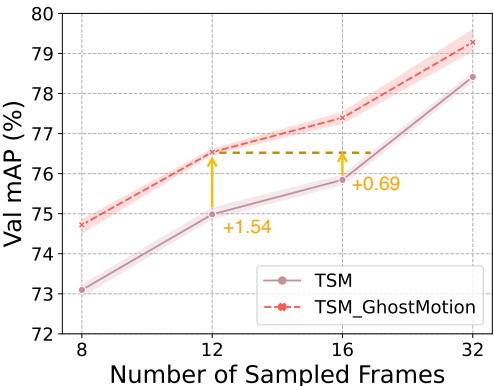

(a) Ghost Motion leads to consistent improvement on Something-Something V1 with different models.

(b) Ghost Motion continuously improves the performance with different sampled frames on ActivityNet.

Figure 1: The performance of Ghost Motion on different methods and datasets. Ghost Motion introduces negligible cost to current competing methods and improves their generalization abilities.

In our work, we propose a data augmentation approach Ghost Motion (GM) which shifts channels along the temporal dimension to propagate motion information to adjacent frames. Specifically, all channels will be shifted for one step along the temporal axis leading to misaligned channel orders, and we interpolate between the original video and the misaligned one to form the new video, named ghost video. It diffuses motion patterns from salient frames into other frames to enhance the overall representative capacity. In this way, we can enforce the model to focus more on the informative frames and prevent it from overfitting non-informative frames to relieve overfitting. Although the shifting operation results in mismatched RGB orders, we surprisingly find the disorder is beneficial to improve generalization as well where it is elaborated in the analysis of Sec. 4.5. Ghost Motion effectively enlarges the input space without crashing the motion pattern and offers continuous variance in the input space which is beneficial to improve the generalization abilities of video recognition methods. Moreover, we find utilizing a hyperparameter Temperature to scale the logits before Softmax can further mitigate overfitting and reduce the calibration error, especially on challenging datasets such as Something-Something V1&V2.

The proposed GM can be easily plugged into existing methods to improve their generalization abilities with a few line codes and brings negligible computational costs. Shown in Fig. 1 (a), GM results in consistent improvements over various 2D models. In Fig 1 (b), GM continuously improves the performance of baseline model with different number of sampled frames. In addition, our method is compatible with existing image-level augmentations. We can jointly apply them to relieve overfitting and further boost the performance. The main contributions are summarized as follow:

- We propose video recognition data augmentation method Ghost Motion (GM) which can *effectively improve the generalization of current video benchmark models* and is *compatible with existing image-level data augmentation approaches*.
- We find smoothing the logits can prevent overconfident predictions to further alleviate overfitting for temporal-dependent datasets such as Something-Something.
- We conduct comprehensive experiments to *validate the strength of Ghost Motion on various datasets and methods*. Extensive ablation with detailed analysis illustrate the motivation and intuition about our GM strategy.

## 2 RELATED WORK

### 2.1 VIDEO RECOGNITION

Video recognition has benefited a lot from the development in image classification, especially the 2D based methods which share the same backbone with the models in image classification. The main focus of these 2D methods lies in temporal modeling. For instance, TSN (Wang et al., 2016) proposes to average the prediction of all the frames to present the final prediction and lays a foundation for later 2D models. While TSM (Lin et al., 2019) is designed to shift parts of the channels among adjacent frames for temporal modeling. Recently, people have paid more attention to multi-scale

temporal modeling and many methods are proposed to not only model the motion variance locally, but capture the global temporal information as well (Li et al., 2020; Wang et al., 2021). Another line of research focuses on using 3D-CNNs to learn the spatial and temporal information jointly, such as C3D (Tran et al., 2015), I3D (Carreira & Zisserman, 2017) and SlowFast (Feichtenhofer et al., 2019). Though being effective, these methods usually cost great computations and are hard to be deployed on edge devices. Based on the structure of Vision Transformers (Dosovitskiy et al., 2020; Liu et al., 2021a), many Transformer-based networks (Fan et al., 2021; Liu et al., 2022; Li et al., 2022) have been proposed recently for spatial-temporal learning in video recognition and shown powerful performance.

Ghost Motion is similar with temporal shift module (TSM) (Lin et al., 2019) as both methods involves the shifting operation and enforces information exchange between neighboring frames. Nevertheless, the differences are: (1) Motivation: TSM is designed as a temporal modeling module which is densely inserted in the network and hard to generalize to other architectures for further improvement; Ghost Motion is a video data augmentation method which manipulates the input data, thus can be generalized to different networks and other image-level data augmentation methods. (2) Specific Designs: TSM shifts limited number of channels at every convolution block and the rest channels are kept in the original place; Ghost Motion shifts all channels initially which causes mismatch in the order of channels and the ghost video will be determined by the sampling coefficient, leading to a enlarged input space.

## 2.2 Mitigating Overfitting

Lots of efforts have been made towards the study of regularization in the image domain to prevent overfitting. Dropout (Srivastava et al., 2014), Weight Decay (Krogh & Hertz, 1991) and Label Smoothing (Szegedy et al., 2016) are widely utilized in image recognition to reduce overfitting. Recently, RMS (Kim et al., 2020) is proposed to vary the magnitude of low-frequency components randomly to regularize the 3D models. Besides these regularization methods, data augmentation methods have also proven to be effective to mitigate the effect of overfitting, especially for datasets which only have limited samples. Traditional augmentation methods, including random flipping, resizing and cropping, are frequently used to enforce invariance in deep networks (He et al., 2016; Huang et al., 2017). Some studies have investigated to enlarge the input space by randomly occluding certain regions in images (DeVries & Taylor, 2017), blending two images (Zhang et al., 2017) or replacing an image patch with the one in another image (Yun et al., 2019). In recent years, automatic augmentation strategies (Cubuk et al., 2018; 2020) are shown to be effective as it considers extensive augmentation types, such as color jittering, shear, translation, etc. However, there has been lack of studies for data augmentation methods in video recognition where overfitting is an even severe problem compared to image classification.

Mixup (Zhang et al., 2017) is a widely used data augmentation method which encourages the model to behave linearly in-between training samples. Ghost Motion is related to mixup as both methods adopt interpolation to generate new training samples. Yet, we emphasize that the two methods enlarge the input space from two perspective and are complementary to each other. Mixup trains the network on convex combinations of pairs of samples and their labels which can be understood as introducing more inter-class data. The success of mixup can be mostly attributed to the linear relationship between training samples and corresponding labels because of the regularization effects it brings (Carratino et al., 2020). Ghost Motion does not touch the label of training sapmples and can be considered as doing interpolation between the original input space and the generated input space which introduces more smoothed intra-class data.

## 3 Methodology

### 3.1 Preliminaries

Given a video $x \in X$ containing T frames $x = \{f_1, f_2, ..., f_T\}$ with its one-hot class label $y \in Y$, we denote $X$ as the input video space, $Y = [0, 1]^K$ and $K$ is the number of classes. A 2D network $F$ maps $x$ to the vector $p = F(x)$ where $p = \{\rho_1, \rho_2, ..., \rho_T\}$ correspond to the predicted logits of T frames. The final prediction $P$ of video $x$ is formed as:

$$P = \mathcal{G}(\rho_1, \rho_2, ..., \rho_T),  \tag{1}$$

where $\mathcal{G}$ is the aggregation function and the most common adapted form is the average function. Incorporated with standard categorical cross-entropy loss, the loss function is defined as:

$$
\begin{aligned}
\mathcal{L} &= -\sum_{k=1}^{K} y_k \log \frac{e^{P_k}}{\sum_{j=1}^{K} e^{P_j}} \\
&= -\sum_{k=1}^{K} y_k \log \frac{e^{\mathcal{G}\left(\rho_{k_1}, \rho_{k_2}, \ldots, \rho_{k_T}\right)}}{\sum_{j=1}^{K} e^{\mathcal{G}\left(\rho_{j_1}, \rho_{j_2}, \ldots, \rho_{j_T}\right)}}.
\end{aligned}
\tag{2}
$$

As $y_k = 1$ and $y_h = 0$ for all $h \neq k$, minimizing this loss function equals maximizing the log-likelihood of the right label. The maximum can hardly be achieved with a finite $P_k$, but it can be approached if $P_k \gg P_h$ which means that the logit corresponding to the correct label is much greater than other logits. With this objective, it may result in overfitting as the model is encouraged to produce overconfident predictions and brings difficulties to generalization. This effect may be even disastrous in video recognition as:

$$
P_k \gg P_h \rightarrow \mathcal{G}\left(\rho_{k_1}, \rho_{k_2}, \ldots, \rho_{k_T}\right) \gg \mathcal{G}\left(\rho_{h_1}, \rho_{h_2}, \ldots, \rho_{h_T}\right),
\tag{3}
$$

which can be achieved more easily if there is an imbalance distribution of $\{\rho_1, \rho_2, \ldots, \rho_T\}$ in each video. Unfortunately, motion-irrelevant frames exists in many video samples. The tendency of giving overconfident predictions may result in large logits of background frames which will overwhelm other frames' predictions and lead to wrong final decision.

There are challenging datasets in video recognition such as Something-Something V1&V2 which involve significant temporal dependency and consists of abundant motion-irrelevant frames. To prevent the final predictions being overwhelmed by background frames, we propose to smooth the logits of each sample on these datasets and we denote it as logits smoothing. This procedure is similar with (Agarwala et al., 2020) where the temperature is tuned delicately to improve the accuracy in image recognition. However, our aim is to prevent overconfident predictions of background frames and relieve the overfitting problem. We introduce a hyperparameter temperature $\tau$ and assign it to each logits, so the standard cross-entropy loss can be redefined as:

$$
\begin{aligned}
\mathcal{L} &= -\sum_{k=1}^{K} y_k \log \frac{e^{\frac{P_k}{\tau}}}{\sum_{j=1}^{K} e^{\frac{P_j}{\tau}}} \\
&= -\sum_{k=1}^{K} y_k \log \frac{e^{\frac{\mathcal{G}\left(\rho_{k_1}, \rho_{k_2}, \ldots, \rho_{k_T}\right)}{\tau}}}{\sum_{j=1}^{K} e^{\frac{\mathcal{G}\left(\rho_{j_1}, \rho_{j_2}, \ldots, \rho_{j_T}\right)}{\tau}}}.
\end{aligned}
\tag{4}
$$

As $\mathcal{G}$ is the average function, this operation equals smoothing the logits of every frame. When $\tau > 1$, the gap between $\frac{P_k}{\tau}$ and $\frac{P_h}{\tau}$ can be reduced which alleviates the overfitting problem. While temperature scaling is commonly used in model calibration (Guo et al., 2017), it only applies temperature during testing for calibration which does not help to improve the generalization of models.

### 3.2 GHOST MOTION

In this section, we introduce our proposed data augmentation method Ghost Motion (GM) which is specially designed for video recognition. Considering a RGB frame $f_i \in \mathbb{R}^{\left(C_i^R, C_i^G, C_i^B\right) \times H \times W}$ with 3 channels in total, we can denote the video sequence as $x \in \mathbb{R}^{T \times \left(C^R, C^G, C^B\right) \times H \times W}$.

Given a random probability $\mu$, we generate a misaligned video $x' \in X'$ by shifting channels for one step along the temporal dimension with a random direction, so that we can represent the generated misaligned frame as:

$$
\begin{cases}
f_i{}' \in \mathbb{R}^{\left(C_i^G, C_i^B, C_{i+1}^R\right) \times H \times W}, & \mu \leq 0.5, \\
f_i{}' \in \mathbb{R}^{\left(C_{i-1}^B, C_i^R, C_i^G\right) \times H \times W}, & \mu > 0.5.
\end{cases}
\tag{5}
$$

With this operation, we can enable information exchange in $x'$. The misaligned frame $f_i{}'$ is composed of three channels, with two channels from the original frame and one from adjacent frame which explicitly facilitate information interaction between neighboring frames. However, directly

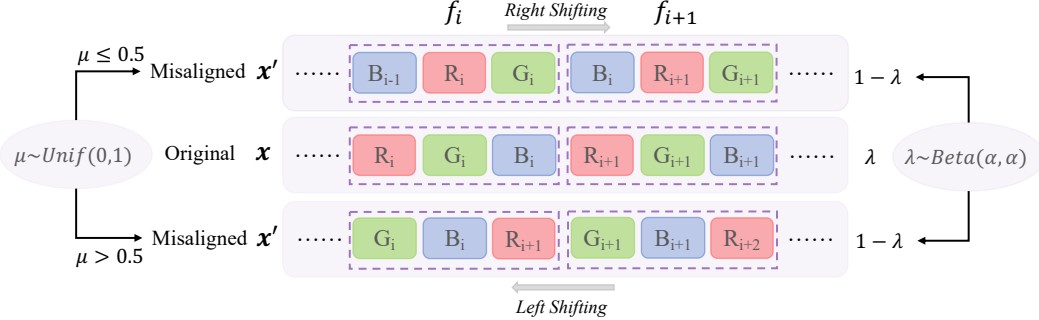

Figure 2: Illustration of Ghost Motion. Given a video $x = \{f_1, f_2, ..., f_T\}$, its channels will be shifted along the temporal dimension to generate a misaligned video $x'$ and the shifting direction is decided by a random probability $\mu$ sampling from uniform distribution. Interpolation beween original video and misaligned video will be implemented to generate the Ghost Video. The coefficient $\lambda$ is sampled from beta distribution to control the degree of interpolation.

using the misaligned video as the input source may not be optimal because the generation follows a fixed pattern and the discrete variance may not be favorable. Therefore, we interpolate between the original video and misaligned video, and sample a coefficient $\lambda$ from the Beta distribution $Beta(\alpha, \alpha)$ to control the strength. In our experiments, we set $\alpha$ to 1 so that $\lambda$ is sampled from the uniform distribution $(0, 1)$ and we can denote the generated training sample $\tilde{x}$ and the corresponding label as:

$$\begin{cases} \tilde{x} = \lambda x + (1 - \lambda) x', \\ \tilde{y} = y. \end{cases} \tag{6}$$

We name $\tilde{x}$ ghost video as there exists artifacts in certain frames. For adjacent frames with great motion variance, the shifting operation will transfer semantic information to other frames, resulting in motion artifacts which looks like the RGB difference data. The reason is motion patterns locates in different spatial regions for neighboring frames and the overlay of channels from different frames will preserve the motion information of both. With this design, the generated data will enforce the model to focus more on the temporal variance guided by the motion artifacts. Besides, motion information may be diffused into non-informative frames and increase their chance to be correctly classified to alleviate the imbalance in aggregation function $\mathcal{G}$. Though the shifting operation leads to mismatch in RGB orders, we find the color alteration is beneficial to further improve the generalization. Note that $x'$ is produced by $x$, so ghost video $\tilde{x}$ shares the same label with original video sample $x$. We denote the generated channels of ghost frame $\tilde{f}_i$ as:

$$\begin{cases} \tilde{C}_i^R = \lambda C_i^R + (1 - \lambda) C_i^G, \\ \tilde{C}_i^G = \lambda C_i^G + (1 - \lambda) C_i^B, \\ \tilde{C}_i^B = \lambda C_i^B + (1 - \lambda) C_{i+1}^R, \end{cases} \mu \le 0.5, \quad \begin{cases} \tilde{C}_i^R = \lambda C_i^R + (1 - \lambda) C_{i-1}^B, \\ \tilde{C}_i^G = \lambda C_i^G + (1 - \lambda) C_i^R, \\ \tilde{C}_i^B = \lambda C_i^B + (1 - \lambda) C_i^G, \end{cases} \mu > 0.5, \tag{7}$$

where $\tilde{C}_i^R, \tilde{C}_i^G, \tilde{C}_i^B$ stand for the RGB channels of the ghost frame $\tilde{f}_i$ respectively. Due to the introduced $\lambda$, the degree of temporal variance can be measured with a linear relation, leading to continuous variance in the input space.

## 4 EMPIRICAL VALIDATION

In this section, we empirically evaluate the performance of the proposed method on various video recognition datasets. We first validate the effectiveness of Ghost Motion (GM) on different models and various datasets. Second, we present the comparisons of our method with competing data augmentation methods in image recognition and prove that GM is compatible with them. Further, we extend GM to other structures and depths of network. Finally, we analyze the effect of logits smoothing, discuss the training cost of different data augmentation methods, and present comprehensive ablation results of design choices and visualizations results.

### 4.1 EXPERIMENTAL SETUP

**Datasets.** Our method is evaluated on six widely-used video datasets. For datasets which contain strong temporal dependency: (1) Something-Something V1&V2 (Goyal et al., 2017) are two hu-

Table 1: Evaluation of Ghost Motion (GM) on Something-Something datasets with different models. The best results are bold-faced. Note that we does not apply random flipping on these two datasets, so the reproduced results of TANet and TDN are lower than the number reported in their papers.

| Method | Sth-Sth V1$^\Theta$ | | | Sth-Sth V2$^\Theta$ | | |
|---|---|---|---|---|---|---|
| | Acc1.(%) | Acc5.(%) | $\Delta$ Acc1.(%) | Acc1.(%) | Acc5.(%) | $\Delta$ Acc1.(%) |
| TSN (Wang et al., 2016) | 17.21 | 43.17 | +1.02 | 30.56 | 61.21 | +1.23 |
| TSN+GM | **18.23** | 44.88 | | **31.79** | 62.52 | |
| TSM (Lin et al., 2019) | 45.34 | 74.45 | +1.54 | 59.14 | 85.60 | +0.69 |
| TSM+GM | **46.88** | 75.76 | | **59.83** | 85.96 | |
| TANet (Liu et al., 2021b) | 45.94 | 75.56 | +1.10 | 59.17 | 85.18 | +0.87 |
| TANet+GM | **47.04** | 76.05 | | **60.04** | 86.25 | |
| TEA (Li et al., 2020) | 48.67 | 77.94 | +0.94 | 61.11 | 87.20 | +0.64 |
| TEA+GM | **49.61** | 78.03 | | **61.75** | 87.04 | |
| TDN (Wang et al., 2021) | 49.69 | 78.74 | +0.82 | 63.10 | 88.12 | +0.71 |
| TDN+GM | **50.51** | 79.06 | | **63.81** | 88.59 | |

Table 2: Evaluation of Ghost Motion (GM) on UCF101, HMDB51, ActivityNet and Mini-Kinetics with TSM (Lin et al., 2019) and TEA (Li et al., 2020). The best results are bold-faced.

| Method | UCF101 | | HMDB51 | | ActivityNet | | Mini-Kinetics | |
|---|---|---|---|---|---|---|---|---|
| | Acc1.(%) | $\Delta$ Acc1.(%) | Acc1.(%) | $\Delta$ Acc1.(%) | mAP(%) | $\Delta$ mAP(%) | Acc1.(%) | $\Delta$ Acc1.(%) |
| TSM | 79.57 | +1.00 | 48.63 | +2.48 | 73.10 | +1.62 | 74.11 | +0.71 |
| TSM+GM | **80.57** | | **51.11** | | **74.72** | | **74.82** | |
| TEA | 80.10 | +1.05 | 49.61 | +3.00 | 73.09 | +1.40 | 75.96 | +0.55 |
| TEA+GM | **81.15** | | **52.61** | | **74.49** | | **76.51** | |

man action datasets which includes 98k and 194k videos respectively. For datasets which require less temporal modeling: (2) ActivityNet-v1.3 (Caba Heilbron et al., 2015) contains 10,024 training videos and 4,926 validation videos with 200 categories and the average duration is 117 seconds; (3) UCF101 (Soomro et al., 2012) dataset originally consists of 13,320 videos with 101 classes and we adopt the first training/testing split for training and evaluation; (4) HMDB51 (Kuehne et al., 2011) dataset is made up of 6,766 videos with 51 classes and we use the original first training/testing split for training and evaluation as well; (5) Kinetics400 (Kay et al., 2017) is a large scale video recognition dataset which is categorized into 400 action classes. Mini-Kinetics is a subset of Kinetics400 with 200 categories in total and we select the classes following (Meng et al., 2020).

**Implementation details.** We uniformly sample 8 frames for each video unless specified. During training, the training data is random cropped to 224 × 224 following (Zolfaghari et al., 2018), and we did not perform random flipping on Something-Something V1&V2. At inference stage, all frames are center-cropped to 224 × 224 and we adopt one-crop one-clip per video during evaluation for efficiency. Similar with CutMix (Yun et al., 2019), we add a hyperparamter $\varrho$ which denotes the probability whether to implement GM for every batch samples. We set $\varrho$ as 0.5 on Something-Something V1&V2 which are temporal-dependent and 0.2 in other datasets. Note that we only apply logits smoothing on Something-Something V1&V2 to calibrate the overconfident predictions and we use the symbol $\Theta$ to denote that this dataset is trained with logits smoothing. Due to limited space, more implementation details can be found in supplementary material.

## 4.2 MAIN RESULTS

**Results on different methods.** We first empirically evaluate the performance of Ghost Motion (GM) on Something-Something V1 and V2 datasets using different models in Tab. 1. It can be observed that GM consistently increases the generalization ability of these methods on both datasets, including the state-of-the-art model TDN (Wang et al., 2021) in 2D methods. The improvements on Sth-Sth V1 are generally larger than those on Sth-Sth V2 as V2 contains more training samples and models will suffer less from overfitting on this dataset. Specifically, the accuracy of TEA (Li et al., 2020) can be improved to 49.61% on Sth-Sth V1 which is similar with our reproduced results of TDN (Wang et al., 2021), adapting more complex structures and taking much longer training time.

**Results on different datasets.** We first validate Ghost Motion on UCF101, HMDB51 and ActivityNet datasets in Tab. 2. The improvements brought by GM are, quite obvious, greater than 1% as the scales of these datasets are relatively small and the overfitting phenomenon is more severe on them. We further implement GM on Mini-Kinetics which is one of the large scale datasets in video

Table 3: Comparisons with competing data augmentation methods on different datasets using TSM (Lin et al., 2019). The best results are bold-faced.

| Method | UCF101 | | HMDB51 | | Sth-Sth V1$^\ominus$ | |
|---|---|---|---|---|---|---|
| | Acc1.(%) | Δ Acc1.(%) | Acc1.(%) | Δ Acc1.(%) | Acc1.(%) | Δ Acc1.(%) |
| TSM | 79.57 | - | 48.63 | - | 45.34 | - |
| TSM+Cutout (DeVries & Taylor, 2017) | 79.04 | −0.53 | 48.69 | +0.09 | 44.68 | −0.66 |
| TSM+CutMix (Yun et al., 2019) | 79.16 | −0.41 | 48.17 | −0.46 | 44.99 | −0.35 |
| TSM+Mixup (Zhang et al., 2017) | 79.83 | +0.26 | 50.39 | +1.76 | 46.03 | +0.69 |
| TSM+AugMix (Hendrycks et al., 2019) | 79.94 | +0.40 | 50.59 | +1.96 | 46.23 | +0.89 |
| TSM+RandAugment (Cubuk et al., 2020) | 82.13 | +2.16 | 52.55 | +3.92 | 47.20 | +1.86 |
| TSM+GM | 80.57 | +1.00 | 51.11 | +2.48 | 46.88 | +1.54 |
| TSM+GM+Mixup | 80.99 | +1.42 | 51.44 | +2.81 | 47.19 | +1.85 |
| TSM+GM+AugMix | 81.10 | +1.53 | 52.68 | +4.05 | 46.93 | +1.59 |
| TSM+GM+RandAugment | **83.24** | **+3.67** | **53.79** | **+5.16** | **48.06** | **+2.72** |

Table 4: Evaluation of Ghost Motion (GM) on Something-Something V1 with different depths.

| Method | Sth-Sth V1$^\ominus$ | |
|---|---|---|
| | Acc1.(%) | Δ Acc1.(%) |
| TSM(R18) | 39.04 | |
| TSM(R18)+GM | **39.80** | +0.76 |
| TSM(R101) | 45.85 | |
| TSM(R101)+GM | **48.05** | +2.20 |

Table 5: Evaluation of Ghost Motion (GM) on Something-Something V1 and Mini-Kinetics with SlowOnly (Feichten-hofer et al., 2019) and Uniformer (Li et al., 2022).

| Method | Sth-Sth V1$^\ominus$ | | Mini-Kinetics | |
|---|---|---|---|---|
| | Acc1.(%) | Δ Acc1.(%) | Acc1.(%) | Δ Acc1.(%) |
| SlowOnly | 41.81 | | 65.29 | |
| SlowOnly+GM | **43.00** | +1.19 | **65.90** | +0.61 |
| Uniformer | 54.45 | | 71.47 | |
| Uniformer+GM | **55.29** | +0.84 | **72.47** | +1.00 |

recognition but GM still increases the generalization ability of the baseline models which validates the effectiveness of our method. Note that Mini-Kinetics has the same number of samples per class with Kinetics400 which means the overfitting phenomenon will not be more severe on this dataset and we conduct experiments on it for efficiency.

### 4.3 COMPARISON WITH OTHER APPROACHES

We compare our method with competing data augmentation methods on UCF101, HMDB51 and Something-Something V1 based on TSM in Tab. 3. It can be observed that popular data augmentation methods like Cutout (DeVries & Taylor, 2017), CutMix (Yun et al., 2019) results in negative effects in video recognition. One possible explanation is that these methods break the motion pattern in video sequences and bring challenges to temporal modeling which is extremely essential in video recognition. Mixup (Zhang et al., 2017) shows consistent improvement on these three datasets as it enforces a regularization effects which is helpful to deal with overfitting. AugMix (Hendrycks et al., 2019) and RandAugment (Cubuk et al., 2020) exhibits great generalization ability as both methods involve multiple augmentation operations and their magnitudes will be adaptively sampled.

Our method outperforms all approaches except for RandAugment, which is composed of various augmentation operations. However, note that GM is compatible with these methods as they only consider image-level transformations and GM mainly focuses on temporal-wise interactions to enlarge the input space. We further combine GM with Mixup, AugMix and RandAugment on these three datasets and the performance can be improved by 3.67%, 5.16% and 2.72% at most.

### 4.4 FURTHER VERIFICATION

**Results on different depths of network.** In previous experiments, all methods are built on ResNet-50 (He et al., 2016) and we further conduct experiments on ResNet-18 and ResNet-101 to verify the effectiveness of our method when evaluated on models with different representation abilities. We can observe from Tab. 4 that GM consistently improves the performance of baseline method and the improvements are increasing (0.76%, 1.54%, 2.20%) with a deeper network. The explanation is that the overfitting phenomenon is more obvious on deeper network as it has stronger representation ability to fit the training data and GM can effectively alleviate this phenomenon which leads to greater improvements in accuracy compared with shallow networks.

**Results on different structures of network.** Although Logits Smoothing and Ghost Motion are inspired by the analysis in 2D-based methods, our method can be implemented on other video recog-

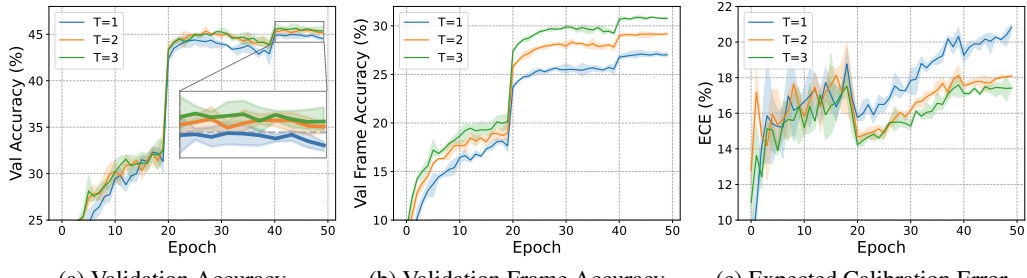

|   |   |   |
|---|---|---|
| (a) Validation Accuracy | (b) Validation Frame Accuracy | (c) Expected Calibration Error |

Figure 3: Validation Accuracy, Validation Frame Accuracy and Expected Calibration Error (ECE) curves of TSM trained with Temperature $T = 1, 2, 3$ on Something-Something V1 dataset.

Table 6: Comparison of Label Smoothing (LAS) and Logits Smoothing (LOS) on Something-Something V1. We set $\alpha = 0.2$ in LAS and $T = 3$ in LOS.

| Method | Acc1.(%) | | | |
|---|---|---|---|---|
| | Baseline | LAS | LOS | LAS+LOS |
| TSM | 45.34 | 45.88 | 45.85 | 46.21 |

Table 7: Training time of different data augmentation methods on Something-Something V1, including Cutout (CU), CutMix (CM), Mixup (MU), Aug-Mix (AM), RandAugment (RA) and GM.

| Method | Training cost (GPU hours) | | | | | | |
|---|---|---|---|---|---|---|---|
| | Baseline | CU | CM | MU | AM | RA | GM |
| TSM | 11 | 12 | 12 | 15 | 13 | 12 | 14 |

nition architecture as it does not touch the structures inside the networks. Therefore, we implement our method on 3D-based method SlowFast (Feichtenhofer et al., 2019) with the slow branch only and Transformer-based method Uniformer (Li et al., 2022) to test the generalization capability of our method. The results in Tab. 5 show that GM leads to improvements in accuracy on both structures and datasets, indicating our method can be applied to other structures to improve their generalization capability as well. The possible explanation is that the data generated by GM has motion artifacts and the explicit pattern in data will enforce the model to focus more on the temporal variance, preventing it from overfitting non-informative frames.

## 4.5 ABLATION AND ANALYSIS

**Effects of logits smoothing.** In this part, we train TSM with $T = 1, 2, 3$ respectively and try to explain what effects logits smoothing brings. We denote Frame Accuracy as the average accuracy of all sampled frames in the training set. Shown in Fig. 3 (a)(b), the model trained with higher temperature has clearly higher frame accuracy which leads to advantages in accuracy of video samples as well. From Fig. 3 (c), one can see that higher temperature results in smaller Expected Calibration Error (ECE) (Guo et al., 2017) which is beneficial to prevent overconfident predictions. Further, we compare logits smoothing with label smoothing (Szegedy et al., 2016) in Tab 6. Results show that both methods lead to increase in accuracy because of the regularization effects and combining two methods can lead to further improvement as they are complementary to each other.

**Training cost.** We measure the training time of different data augmentation methods with the same training setting in Tab 7. Theoretically, these should bring negligible extra computational costs, but we observe time increases in all methods. Mixup costs the most time which means the interpolation of raw data and the manipulation of labels will increase wall-clock time despite their theoretical efficiency. The training time of Cutout and CutMix are less as they only process a small crop of the raw data compared to Mixup and GM, which need to do interpolation on the whole data. Though AugMix and RandAugment involve multiple augmentation operations, these transformations are traditional data augmentation techniques which can be efficiently implemented with small time increases. We hope these analysis can provide insights for later works.

**Design choices.** In this part, we first empirically compare different design choices of Ghost Motion (GM) in Tab. 8. At first, we explore other selections of hyperparameter $\alpha$ which control the beta distribution and the results show that $\alpha = 1$ results in the best performance. Besides, we compare GM with its variants which shift channels for two step and shift for single direction. The two-step shift obtains little worse results than one-step shifts and the possible reason is that two-step shift will lead to an offset in frame location. GM outperforms one-way shift obviously as it offers more variance in data which is helpful to improve the generalization of baseline method. Moreover, we implement GM in the feature space following (Verma et al., 2019) which applies Mixup in the

intermediate features. However, results show that this measure leads to negative effects compared with GM and we consider the misaligned channel order caused by GM in the feature space may be one of the possible reasons.

As implementing GM leads to the stress of the motion information and alteration of the video color, we compare GM with two methods: (1) Motion Exchange(ME): shifting channels along temporal dimension to enforce information exchange but keep the original RGB order; (2) Color Jittering(CJ): randomly changing the brightness, contrast, saturation and hue of the video. We can observe both ME and CJ lead to increase in accuracy and the improvement that GM results in is bigger than the increase brought by ME+CJ which means the combination of the two effects can boost the performance of each other. Another important design of GM is the interpolation between the origi-

Table 8: Ablation of design choices of Ghost Motion on ActivityNet. The best results are bold-faced and mAPs are averaged over multiple runs.

| Method | Specification | mAP(%) |
|---|---|---|
| TSM (Lin et al., 2019) | - | 73.10±0.14 |
| TSM+GM | $Beta(0.5, 0.5)$ | 74.35±0.09 |
| TSM+GM | $Beta(2.0, 2.0)$ | 73.98±0.07 |
| TSM+GM | Two-Step Shift | 74.33±0.22 |
| TSM+GM | One-Way Shift | 73.91±0.03 |
| TSM+GM | Manifold | 73.22±0.15 |
| TSM+ME | Motion Exchange | 73.79±0.10 |
| TSM+CJ | Color Jittering | 73.89±0.25 |
| TSM+GM | w/o interpolation | 74.03±0.05 |
| TSM+GM | - | **74.72±0.04** |

nal video and the misaligned video. Therefore, we compare GM with its variant which directly using the misaligned video as the input. The performance of GM without interpolation is lower than our method which demonstrates the superiority of this design as it offers a continuous distribution of the input data which enlarges the input space and is beneficial for classification.

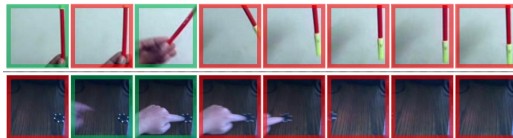
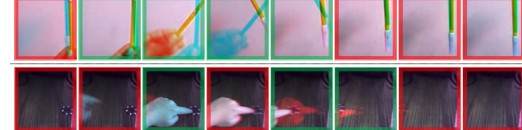

(a) Frame-wise predictions of TSM trained with standard protocol. Original frames are shown.

(b) Frame-wise predictions of TSM trained with Ghost Motion. Frames generated by GM are shown.

Figure 4: Visualization results on Something-Something V1 dataset. Frames annotated with red rectangles are wrongly classified and the green ones represent correctly categorized frames.

**Visualizations.** We visualize the frame-wise predictions of TSM in Fig. 4. The left two samples are wrongly classified by the model trained with standard protocol as the final predictions are overwhelmed by motion-irrelevant frames. In contrast, the right samples are correctly classified by the model trained with GM and there exists motion artifacts in some frames as each frame exchanges parts of the channels with adjacent frames. The motion artifacts looks like the RGB difference between two consecutive frames which describe the appearance change. However, compared to RGB difference, the data generated by GM does not loss information on the static regions as we only shift parts of the channels rather than subtracting the similar parts. Therefore, the data generated by GM could be seen as a new modality which restore the good attributes of RGB and RGB difference. As a result, the model trained with GM can focus more on motion frames guided by the motion artifacts, which prevents the model from overfitting background frames towards the ground truth label and results in better generalization ability.

## 5 CONCLUSION

This paper probes into the overfitting problem in video recognition and proposes a special designed data augmentation method Ghost Motion (GM). Specifically, GM shifts channels along the temporal dimension to enforce motion-relevant information exchange among neighboring frames. This can enhance the learning process as motion information can be diffused into other frames, resulting in motion artifacts which emphasize the motion-relevant frames. By doing so, GM can prevent the model from overfitting non-informative frames and increase its generalization ability for better performance. Comprehensive experiments on popular datasets and methods demonstrate the effectiveness and generalization of GM. Moreover, GM is compatible with existing image-level data augmentation approaches to further improve the performance of video recognition methods.

ETHICS STATEMENT

In our paper, we strictly follow the ICLR ethical research standards and laws. To the best of our knowledge, our work abides by the General Ethical Principles.

REPRODUCIBILITY STATEMENT

We adhere to ICLR reproducibility standards and ensure the reproducibility of our work. All datasets we employed are publicly available. As we have included comprehensive empirical validations on various methods and datasets in our work, we provide corresponding training setting for our method in the appendix. Moreover, we will provide the code to reviewers and area chairs using an anonymous link when the discussion forums is open.

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

## A APPENDIX

### A.1 DATA PRE-PROCESSING

We uniformly sample 8 frames for each video on all datasets, except the experiment over Uniformer (Li et al., 2022) on Something-Something V1 dataset where we sample 16 frames for every video. During training, the training data is randomly cropped to $224 \times 224$ following (Lin et al., 2019), and we perform random flipping except for Something-Something (Goyal et al., 2017) datasets. At inference stage, all frames are center-cropped to $224 \times 224$, obtaining a one-crop one-clip per video for evaluation, except the experiment over Uniformer (Li et al., 2022) on Something-Something V1 dataset where we adopt three-crop for evaluation.

### A.2 IMPLEMENTATION DETAILS

All results reported in our paper are run on NVIDIA Tesla V100 GPUs with mixed precision. All the models are pretrained on ImageNet except for the experiment of Uniformer (Li et al., 2022) on Something-Something V1 which utilize the pre-trained model on Kinetics400 following the original setting. We reimplement all 2D methods in our code and use the official implementations for Slowonly (Feichtenhofer et al., 2019)[1] and Uniformer[2].

For Ghost Motion, $\alpha$ is set to 1 in all our experiments and we change the hyperparamter $\varrho$ which denotes the probability whether to implement GM for every batch samples on different datasets. Note that we only apply Logits Smoothing to temporal-dependent datasets such as Something-Something V1&V2.

#### A.2.1 TEMPORAL SENSITIVE DATASETS

We set $\varrho$ to 0.5 to enable more variance in training samples on Something-Something V1&V2 as the overfitting phenomenon is the most severe on these two datasets. For Logits Smoothing, the temperature is set to 3,2 for Something-Something V1&V2, respectively.

#### A.2.2 TEMPORAL INSENSITIVE DATASETS

We set $\varrho$ to 0.2 for Ghost Motion on Mini-Kinetics (Kay et al., 2017), ActivityNet (Caba Heilbron et al., 2015), HMDB51 (Kuehne et al., 2011) and UCF101 (Soomro et al., 2012) and we did not apply Logits Smoothing on these datasets.

---

[1]https://github.com/facebookresearch/SlowFast
[2]https://github.com/Sense-X/UniFormer/tree/main/video$_{c}lassification$

