# OpenReview forum: "Probing into Overfitting for Video Recognition"
_ICLR.cc/2023/Conference — Submitted to ICLR 2023_

### Official Review · Reviewer_Xyj5 · 2022-10-17

**Confidence:** 4
**Correctness:** 3
**Technical Novelty And Significance:** 3
**Empirical Novelty And Significance:** 3
**Recommendation:** 6

**Clarity, Quality, Novelty And Reproducibility:**

Clarity: The citing format is not precise. Many citations in the paper are supposed to be ~\citep{} instead of ~\cite{}. For example, in the first paragraph of the introduction: TSM Lin et al. (2019) ---> TSM (Lin et al. 2019). Abundant wrong citation formats distract the clarity of the paper.

Reproducibility: The implementation details are specific and easy to follow for the practitioners.

Originality: The method is novel and original, though the shifting trick is introduced in the video action recognition field before [1].

[1] TSM: Temporal Shift Module for Efficient Video Understanding. ICCV 2019

**Strength And Weaknesses:**

Strength:
+ The way of combining channel disorder and temporal misalignment is novel and interesting. This approach enlarges the input space and transfers the semantic frames to adjacent frames.
+ The method is computationally friendly and introduces minimal overhead.
+ The experimental results are comprehensive on various benchmarks including SSv1, SSv2, UCF, HMDB, ActivityNet and Mini-Kinetics.
+ The paper is generally easy to follow.

Weakness:
- Miss the ablation study on the disorder channel, which is one of the key components in the paper. Namely, what the performance will be if the mixing video is TSM-like, $(C^R_{i}, C^G_{i}, C^B_{i+1})\times H \times W$ or $(C^R_{i-1}, C^G_{i}, C^B_{i})\times H \times W$.
- All the baseline performance is far lower than SOTA. I understand for a fair comparison, the paper unifies the training and test recipes and reproduces the 'bad' results of many previous methods. But the community would be more interested in how much the augmentation technique pushes prior arts and the absolute improvement on top of the baseline methods.
- The logits smoothing is orthogonal to the ghost motion augmentation and more looks like add-on to improve the performance on Something-Something dataset. What is the result only equipped with ghost motion augmentation? Is it still effective?

**Summary Of The Paper:**

The paper proposes an augmentation technique for video action recognition named GM (ghost motion). In specific, GM generates the fused clips via the shifted channel and temporal misalignment, encouraging the model to alleviate over-fitting. Besides, the authors apply logits smoothing on the temporal-sensitive benchmark (SSv1/v2) for emphasizing temporal dependency. The experiments demonstrate that GM improves frame-wise accuracy and boosts multiple baseline architectures significantly.

**Summary Of The Review:**

Overall, it is a good paper. The paper introduces a simple yet effective augmentation method to resist over-fitting for video action recognition. The experiments verify the augmentation boosts the performance considerably across the board. Thus, my initial rating is borderline accept. I would like to see the responses from the authors during the discussion period.

---

> ### Author Response · Authors · 2022-11-15
> **Response to Reviewer Xyj5**
>
> We appreciate the Reviewer’s approval and constructive suggestions for us to improve our work. We make the response as below.
>
> **Weakness 1: Ablation on the disorder channel.**
> Thanks for the great suggestion and we really appreciate it for carefully reading our paper.
> Indeed, implementing GM can lead to two effects: (1) stress of the motion information; (2) alteration of the video color.
> Hence, we compare GM with two baseline methods on ActivityNet for ablation: (1) Motion Exchange(ME): shifting channels along temporal dimension to enforce information exchange but keep the original RGB order of all frames (disorder channel); (2) Color Jittering(CJ): randomly changing the brightness, contrast, saturation and hue of the video (shown in Tab 6).
> | Method | mAP |
> | :-----: | :-----: |
> | TSM | 73.10% |
> | TSM+ME | 73.79%(+0.69) |
> | TSM+CJ | 73.89%(+0.79) |
> | TSM+GM | 74.72%(+1.62) |
> | |
> | TEA | 73.09% |
> | TEA+ME | 73.57%(+0.48) |
> | TEA+CJ | 73.54%(+0.45) |
> | TEA+GM | 74.49%(+1.40) |
>
> Results show that Motion Exchange leads to improvement of baseline methods as it will enforce the model to focus more on motion artifacts instead of classifying based on static appearance. Further, the improvement that GM results in is bigger than the increase brought by ME, CJ and ME+CJ which means the combination of the two effects can boost the performance of each other. Thanks again for the advice and we will add this part in our revision.
>
>
> **Weakness 2: Results lower than SOTA.**
> Thanks for asking this question. As there are too many experiments in our paper to validate GM on various methods and datasets, we sample 8 frames for each video and adopt one-crop one-clip per video during evaluation for efficiency. Note that we have demonstrated GM can consistently improve the performance of baseline method with different depths (see Tab 4) and sampled frames (see Fig 1.(b)) which proves the generalization ability of GM on models with different representation abilities. Further, we conduct experiments over competing method Uniformer[3] with more sampled frames and adopt multi-crop evaluation on Something-Something V1 dataset:
> | Method  | Top-1 Acc. |
> | :-----: | ------------ |
> | Uniformer | 54.45% |
> | Uniformer+GM | 55.29%(+0.84) |
>
> One can observe that Uniformer obtains much stronger performance compared with the result in paper (41.74%) and GM still improves the accuracy by 0.84% which validates its efficacy. We admit that this is not the SOTA result in video recognition, but we have tried our best to provide results with the highest accuracy based on our computation resources.
>
>
> **Weakness 3: Results only with GM.**
> We thank the reviewer for the good question. Indeed, logits smoothing is orthogonal to GM and we add it on Something-Something datasets as we found it not only results in lower Expected Calibration Error, but slightly increases the validation accuracy of these methods. We further conduct experiments over TSM[1]/TANet[4]/TEA[2]/TDN[5] without logits smoothing on Something-Something V1 dataset:
> | Method  | Top-1 Acc. | Top-1 Acc.(GM) |  Top-1 Acc.(GM+smoothing) |
> | :-----: | ------------ | :-----: | :-----: |
> | TSM | 45.34% | 46.48%(+1.14) | 46.88%(+1.54) |
> | TANet | 45.94% | 46.93%(+0.99) | 47.04%(+1.10) |
> | TEA | 48.67% | 49.51%(+0.84) | 49.61%(+0.94) |
> | TDN | 49.69% | 50.32%(+0.63) | 50.51%(+0.82) |
>
> It can be seen that GM+smoothing obtains the highest accuracy among the three columns, though GM is still effective as it contributes most of the improvements over baseline methods.
>
>
> **Clarity: Citing format.**
> Thanks for pointing it out and sorry for the confusion brought in reading. We will update the revision with the precise citing format in the next few days.
>
>
>
> Again, we thank the reviewer for the valuable advice for us to improve the paper. We are actively available until the end of this rebuttal period and let us know if you have any further questions. Looking forward to hearing back from you.
>
>
> [1] Lin J, Gan C, Han S. Tsm: Temporal shift module for efficient video understanding[C]//Proceedings of the IEEE/CVF International Conference on Computer Vision. 2019: 7083-7093.
> [2] Li Y, Ji B, Shi X, et al. Tea: Temporal excitation and aggregation for action recognition[C]//Proceedings of the IEEE/CVF conference on computer vision and pattern recognition. 2020: 909-918.
> [3] Li K, Wang Y, Zhang J, et al. Uniformer: Unifying convolution and self-attention for visual recognition[J]. arXiv preprint arXiv:2201.09450, 2022.
> [4] Liu Z, Wang L, Wu W, et al. Tam: Temporal adaptive module for video recognition[C]//Proceedings of the IEEE/CVF International Conference on Computer Vision. 2021: 13708-13718.
> [5] Wang L, Tong Z, Ji B, et al. Tdn: Temporal difference networks for efficient action recognition[C]//Proceedings of the IEEE/CVF Conference on Computer Vision and Pattern Recognition. 2021: 1895-1904.

---

> ### Author Response · Authors · 2022-11-17
> **Sincerely looking forward to your further feedback**
>
> Dear Reviewer Xyj5,
>
> Thanks for your valuable comments made in the review process. As we have promised, we have revised the draft based on the suggestions from all Reviewers, including the following changes :
> ***
> * de-emphasized logtis smoothing and compared it with label smoothing[1];
> * added the discussion of training cost of existing data augmentation methods;
> * included stronger results of Uniformer[2] on Something-Something V1 dataset;
> * added ablation of Motion Exchange (ME);
> * revised the citing format;
> * revised the writing of certain descriptions;
> * corrected the typos.
>
> [1] Szegedy C, Vanhoucke V, Ioffe S, et al. Rethinking the inception architecture for computer vision[C]//Proceedings of the IEEE conference on computer vision and pattern recognition. 2016: 2818-2826.
> [2] Li K, Wang Y, Zhang J, et al. Uniformer: Unifying convolution and self-attention for visual recognition[J]. arXiv preprint arXiv:2201.09450, 2022.
> ***
>
> We have tried our best to upload this draft as early as we can and please let us know if you have further comments so that we still have a chance to improve our paper before the deadline. Given the discussion deadline is approaching, we really hope to have a further discussion with you to see if our responses solve your concerns. Thank you for being with us so far.
>
>
> Sincerely,
> Authors

---

> > ### Comment · Reviewer_Xyj5 · 2022-11-17
> > **Response to rebuttal**
> >
> > I appreciate the efforts the authors put in during the rebuttal. Most of my concerns are well-addressed by the response. I would like to keep my score.

---

### Official Review · Reviewer_6srd · 2022-10-24

**Confidence:** 4
**Correctness:** 3
**Technical Novelty And Significance:** 2
**Empirical Novelty And Significance:** 3
**Recommendation:** 5

**Clarity, Quality, Novelty And Reproducibility:**

The method is presented with clarity, which adds to the reproducibility of the work, though there are some confusions in the experimental results. I am less confident in originality as the proposed techniques are either well studied in prior work (legit smoothing) or simply adapted from related problems (channel-wise temporal shifts).

**Strength And Weaknesses:**

## Pros

- This paper studies an important problem of video data augmentation, where there has been limited prior work in the literature.
- The proposed method is simple and appears effective across datasets and models.
- Ablation experiments on the augmentation parameters are provided.

## Cons

- Methods
	- I don't feel that the motivation for GM is strong enough. Specifically, how does the information transfer between frames lead to stronger temporal modeling at test time (when the channels are not shifted)? It would be nice to have experiments supporting that models trained with GM is more sensitive to temporal cues than spatial ones.
	- I am also not sure about the design choice, where only red or blue channels are shifted, but not the green channel. It would be nice if more explanations or justifications are provided. It would also be interesting to see if shifting all channels with interpolation can achieve similar effects.
	- The technical contribution from this work is somewhat limited: GM mirrors the temporal shift operation of TSM in the input image space, with similar objectives (improve temporal modeling); temperature scaling has been widely used in model calibration literatures to reduce overfitting, and should not be considered an original contribution from this work.

- Experiments
	- The effect of logit smoothing and ghost motion is not well disentangled in the experimental results. From figure 3 it appears that temperature tuning alone improves accuracy by ~1%. Is this the main source of improvements in the main tables, or do the gains come from the GM augmentation instead?
	- The improvements over existing image-based augmentations is not particularly significant (e.g. <1% over MixUp).
	- There does not seem to be comparison to prior video data augmentation methods such as VideoMix (https://arxiv.org/abs/2012.03457).

- Minor comments
	- I assume that the effectiveness of method depends on the video frame rate. If frame sampling is dense enough, adjacent frames will be too similar that shifting does not make a significant difference. Maybe it would be nice to include additional discussion/experiments on this.

**Summary Of The Paper:**

This paper makes two contributions to reduce overfitting of video recognition models: Ghost Motion (GM) data augmentation which randomly shifts one RGB channel forward or backward in time, and logit smoothing (temperature scaling) to reduce overconfidence on background frames. The authors hypothesize that GM works by exchanging information between neighboring frames, creating motion artifacts that guide the model to focus on temporal cues. Experiments on multiple video action datasets show the proposed training scheme reduces overfitting and improves test accuracy of classifiers.

**Summary Of The Review:**

Although this work has its strengths, I lean toward rejection of the paper at its current state, due to lack of technical contributions and empirical justifications of some claims made in the paper, as detailed in the weaknesses above. I look forward to the authors' response to resolve any confusions/misunderstandings.

UPDATE: Increased rating to borderline reject post-rebuttal.

---

> ### Author Response · Authors · 2022-11-15
> **Response to Reviewer 6srd(Part 1)**
>
> We appreciate the Reviewer's feedback. We make further explanations to clarify the Reviewer's concerns based on several key points as below.
>
> **Weakness 1: Motivation not strong enough.**
> Thanks for asking this question. Our work is built on the observation that 2D networks average the logits of all frames to vote for a final decision and the models are likely to overfit background frames with less semantic information which is detrimental for temporal modeling. The core idea of GM is to enlarge the input space during training by introducing misaligned videos with disordered channels so that we can enforce the model to focus more on motion frames guided by the motion artifacts which we regarded as 'beneficial for temporal modeling' during training (shown in analysis of Fig 4).
> Note that our training set is made of original video, misaligned video and ghost video which is the interpolation between these two, and we think enforcing the model to fit different motion patterns will help the model to learn more generalized representations which will result in improvement at test time when the channels are not shifted.
>
> **Weakness 2: Explanations of design choices.**
> We thank the Reviewer for this comment. However, we want to clarify that our design is
> shifting all the channels by one step instead of only shifting red and blue channels.
> For example, given a video with $T$ frames, we first reshape the dimension from $T \times 3 \times H \times W$ to $(T \times 3) \times H \times W$ so that we will have $(T \times 3)$ channels in total.  We denote the RGB channels of the $i$ frame in original video as: $ (C_{i}^{R}, C_{i}^{G},, C_{i}^{B},) $ and we shift all channels by one step with a random direction,
> the frame after shifting will be: $(C_{i-1}^{B}, C_{i}^{R}, C_{i}^{G})$ or $(C_{i}^{G}, C_{i}^{B}, C_{i+1}^{R})$ which corresponds to right shifting and left shifting respectively (i.e., channels $i, i+1, i+2$ will be in the location of $i+1, i+2, i+3$ with right shifting).
>
> This behavior results in two effects: (1) stress of the motion information because channels from adjacent frames are shifted to current frame; (2) alteration of the video color because of the mismatch of RGB order. Based on our experiments in reply to Weakness 1 pointed Reviewer Xyj5, both effects help to improve the generalization of baseline method and the improvement that GM results in is bigger than the increase brought by the sum of the two effects which demonstrates the effectiveness of our design.
>
> **Weakness 3: Technical contribution limited.**
> Thanks for this question.
> As for GM, we have discussed the difference with TSM[1] in Related Work as:
> (1) Motivation: TSM is considered as a temporal modeling module which is densely inserted into the network and hard to generalize to other architectures for further improvement. GM is designed to prevent overfitting which manipulates the input data and can be generalized to different methods.
> (2) Specific Design: TSM only shifts a small portion of channels and the rest channels are kept in the original location. While GM will first reshape Channel ($C$) and Temporal ($T$) dimension into one dimension ($C \times T$), then shift all channels by one step to cause the mismatch in channel orders which brings motion exchange and color variance to alleviate overfitting.
>
> As for logits smoothing, it is not our main contribution and indeed, temperature scaling[2] has been widely used to reduce ECE for calibration. However, they only apply the temperature during testing which does not help to improve the generalization of models. In contrast, we apply logits smoothing both in training and testing, and we found it not only results in lower ECE, but slightly increases the validation accuracy of these methods.
>
> Further, apart from the differences in method, we want to emphasize that the overfitting problem in video recognition has been seldom touched and it is a more severe problem compared to image recognition which needs more attention. Starting from the analysis on 2D networks, our method made an attempt to design a specialized data augmentation for video recognition which has shown effectiveness on various architectures (2D/3D/Transformer) and datasets, and our work is compatible with current image-level data augmentation methods to further improve the generalization of video recognition models.

---

> ### Author Response · Authors · 2022-11-15
> **Response to Reviewer 6srd(Part 2)**
>
> **Weakness 4: Entangled experiments results.**
> Thanks for pointing it out. First, we give the results of TSM with different temperatures on Something-Something V1 dataset:
> | Method | Top-1 Acc. |
> | :-----: | :-----: |
> | TSM(T=1) | 45.34% |
> | TSM(T=2) | 45.77%(+0.43) |
> | TSM(T=3) | 45.85%(+0.51) |
>
> We can observe logits smoothing slightly increases the validation accuracy of these methods, but the improvements are actually less than 1%. We further conduct experiments over TSM[1]/TANet[3]/TEA[4]/TDN[5] without logits smoothing on Something-Something V1 dataset:
> | Method  | Top-1 Acc. | Top-1 Acc.(GM) |  Top-1 Acc.(GM+smoothing) |
> | :-----: | ------------ | :-----: | :-----: |
> | TSM | 45.34% | 46.48%(+1.14) | 46.88%(+1.54) |
> | TANet | 45.94% | 46.93%(+0.99) | 47.04%(+1.10) |
> | TEA | 48.67% | 49.51%(+0.84) | 49.61%(+0.94) |
> | TDN | 49.69% | 50.32%(+0.63) | 50.51%(+0.82) |
>
> It can be seen that GM+smoothing obtains the highest accuracy among the three columns, though GM is still effective as it contributes most of the improvements over baseline methods.
>
> **Weakness 5: Improvement not significant.**
> We compare GM with image-level data augmentation methods in Tab 3, and GM outperforms all methods except RandAugment[6], which is composed of multiple augmentation operations. Nevertheless, we stress that our method is compatible with existing image-level data augmentation methods and we can easily build on top of other approaches to further enhance the performance. Here we list the improvements of GM when building on top of Mixup[7], AugMix[8], RandAugment[6]:
> | Method  | Top-1 Acc.(UCF101) | Top-1 Acc.(HMDB51) |  Top-1 Acc.(Sth-Sth V1) |
> | :-----: | ------------ | :-----: | :-----: |
> | TSM | 79.57% | 48.63% | 45.34% |
> | |
> | TSM+Mixup | 79.83% | 50.39% | 46.03% |
> | TSM+Mixup+GM | 80.99%(+1.16) | 51.44%(+1.05) | 47.19%(+1.16) |
> | |
> | TSM+AugMix | 79.94% | 50.59% | 46.23% |
> | TSM+AugMix+GM | 81.10%(+1.16) | 52.68%(+2.09) | 46.93%(+0.70) |
> | |
> | TSM+RangAugment | 82.13% | 52.55% | 47.20% |
> | TSM+RangAugment+GM | 83.24%(+1.11) | 53.79%(+1.24) | 48.06%(+0.86) |
>
> One can observe very consistent improvements when building GM on top of different image-level augmentations which proves the effectiveness and generalization of our design.
>
> **Weakness 6: Comparison with VideoMix.**
> We thank the Reviewer for reminding us about this work[9]. Similar with the experimental setting in Tab 3, we conduct experiments on three datasets with different hyperparameters (adopting from their codes) to compare VideoMix with GM and the results are shown in the table:
> | Method  | Top-1 Acc.(UCF101) | Top-1 Acc.(HMDB51) |  Top-1 Acc.(Sth-Sth V1) |
> | :-----: | ------------ | :-----: | :-----: |
> | TSM | 79.57% | 48.63% | 45.34% |
> | TSM+VideoMix($\alpha$=0.4) | 78.62%(-0.95) | 47.06%(-1.57) | 45.62%(+0.28) |
> | TSM+VideoMix($\alpha$=2.0) | 78.24%(-1.33) | 48.11%(-0.52) | 45.25%(-0.09) |
> | TSM+VideoMix($\alpha$=8.0) | 77.93%(-1.64) | 48.92%(+0.29) | 44.48%(-0.86) |
> | TSM+GM | 80.57%(+1.00) | 51.11%(+2.48) | 46.88%(+1.54) |
>
> We find VideoMix seems to fail on 2D network and results in even worse performance compared with baseline model. This finding aligns with our results in Tab 3 where CutMix leads to worse results across three datasets and VideoMix seems an extension of CutMix on video recognition.

---

> ### Author Response · Authors · 2022-11-15
> **Response to Reviewer 6srd(Part 3)**
>
> **Weakness 7: Dense frame sampling.**
> Thanks for the good question. In Fig 1.(b), we have included the results of GM with different numbers of sampled frames on ActivityNet and we list the results in this table:
>
> | Method | Frame | mAP |
> | :-----: | :-----: |  :-----: |
> | TSM | 8 | 73.10% |
> | TSM+GM | 8 | 74.72%(+1.62) |
> | TSM | 12 | 74.98% |
> | TSM+GM | 12 | 76.54%(+1.56) |
> | TSM | 16 | 75.84% |
> | TSM+GM | 16 | 77.40%(+1.56) |
> | TSM | 32 | 78.41% |
> | TSM+GM | 32 | 79.28%(+0.87) |
>
> The results show that GM consistently improves the performance of baseline model with different sampled frames and there is 0.87% improvement even with 32 sampled frames. Consider the extreme case when we sample enough frames and adjacent frames will be very similar or even the same, GM will degenerate to Color Jittering as there will be no information exchange in adjacent frames but the color of the video will vary because of the mismatch in RGB order. Based on our experiments in reply to Weakness 1 pointed Reviewer Xyj5, Color Jittering can still improve the generalization of baseline method which means GM will not lose efficacy even in this extreme case. Note that GM is effective in most scenarios as sampling too many frames will bring unaffordable computational costs.
>
>
> We thank the Reviewer again for the suggestions which help us to improve the work. We are actively available until the end of this rebuttal period and let us know if you have any further questions. Looking forward to hearing back from you.
>
>
> [1] Lin J, Gan C, Han S. Tsm: Temporal shift module for efficient video understanding[C]//Proceedings of the IEEE/CVF International Conference on Computer Vision. 2019: 7083-7093.
> [2] Guo C, Pleiss G, Sun Y, et al. On calibration of modern neural networks[C]//International conference on machine learning. PMLR, 2017: 1321-1330.
> [3] Liu Z, Wang L, Wu W, et al. Tam: Temporal adaptive module for video recognition[C]//Proceedings of the IEEE/CVF International Conference on Computer Vision. 2021: 13708-13718.
> [4] Li Y, Ji B, Shi X, et al. Tea: Temporal excitation and aggregation for action recognition[C]//Proceedings of the IEEE/CVF conference on computer vision and pattern recognition. 2020: 909-918.
> [5] Wang L, Tong Z, Ji B, et al. Tdn: Temporal difference networks for efficient action recognition[C]//Proceedings of the IEEE/CVF Conference on Computer Vision and Pattern Recognition. 2021: 1895-1904.
> [6] Cubuk E D, Zoph B, Shlens J, et al. Randaugment: Practical automated data augmentation with a reduced search space[C]//Proceedings of the IEEE/CVF conference on computer vision and pattern recognition workshops. 2020: 702-703.
> [7] Zhang H, Cisse M, Dauphin Y N, et al. mixup: Beyond empirical risk minimization[J]. arXiv preprint arXiv:1710.09412, 2017.
> [8] Hendrycks D, Mu N, Cubuk E D, et al. Augmix: A simple data processing method to improve robustness and uncertainty[J]. arXiv preprint arXiv:1912.02781, 2019.
> [9] Yun S, Oh S J, Heo B, et al. Videomix: Rethinking data augmentation for video classification[J]. arXiv preprint arXiv:2012.03457, 2020.

---

> ### Author Response · Authors · 2022-11-17
> **Sincerely looking forward to your further feedback**
>
> Dear Reviewer 6srd,
>
> Thanks for your valuable comments made in the review process. As we have promised, we have revised the draft based on the suggestions from all Reviewers, including the following changes :
> ***
> * de-emphasized logtis smoothing and compared it with label smoothing[1];
> * added the discussion of training cost of existing data augmentation methods;
> * included stronger results of Uniformer[2] on Something-Something V1 dataset;
> * added ablation of Motion Exchange (ME);
> * revised the citing format;
> * revised the writing of certain descriptions;
> * corrected the typos.
>
> [1] Szegedy C, Vanhoucke V, Ioffe S, et al. Rethinking the inception architecture for computer vision[C]//Proceedings of the IEEE conference on computer vision and pattern recognition. 2016: 2818-2826.
> [2] Li K, Wang Y, Zhang J, et al. Uniformer: Unifying convolution and self-attention for visual recognition[J]. arXiv preprint arXiv:2201.09450, 2022.
> ***
>
> We have tried our best to upload this draft as early as we can and please let us know if you have further comments so that we still have a chance to improve our paper before the deadline. Given the discussion deadline is approaching, we really hope to have a further discussion with you to see if our responses solve your concerns. Thank you for being with us so far.
>
>
> Sincerely,
> Authors

---

> > ### Comment · Reviewer_6srd · 2022-11-22
> > **Response to Rebuttal**
> >
> > I would like to thank the authors for the elaborated response to the initial review. I appreciate the clarifications on the motivation and procedure of GM, as well as the additional ablation studies that provide further insight into its mechanism. I will increase my rating for the new revision.
> >
> > However, there are still a few concerns that make it difficult for me to recommend acceptance yet:
> > - When disentangled from logic smoothing, the improvements from GM is not significant (~1% on SS v1), which might not justify the 30% longer training time. Being compatible with other augmentation methods is a plus, though.
> > - I am still not fully convinced that the improvements come from temporal modeling. If this is indeed the case, we should expect to see a larger gain on SS v1/v2 (temporally heavy dataset) than UCF and HMDB (spatially heavy), which does not seem to be the case. If this is the main claim of the paper, there should be at least some temporal focused analysis such as testing with shuffled frames and comparing accuracy vs normal frame order.
> > - As other reviewers have also pointed out, logit smoothing is not novel and somewhat distracts from the prime focus. The paper should probably concentrate on the GM data augmentation entirely.
> >
> > Minor comments:
> > - I think the comparison of training time is helpful but would prefer if the authors included the GPU model as well, and additionally mention if the numbers are for 1 epoch or full training.
> > - Results on SOTA architectures are a bit limited, though this is understandable under constraints in training resources. The authors should include the exact model names within each model class (e.g. SlowOnly 4x16/8x8; UniFormer-S/B/L) when presenting the results.

---

> > > ### Author Response · Authors · 2022-11-23
> > > **Thank you! Further response(Part 1)**
> > >
> > > Dear Reviewer 6srd,
> > >
> > > Thank you very much for your thorough reply and your recognition for our effort in the rebuttal.
> > >
> > > For your remaining concerns, we would like to make some further explanations and expect that will help for a better understanding of our work:
> > >
> > >
> > > **Concern1: Results not significant.**
> > > First, we want to stress that GM is a plug-in strategy and our comprehensive experiments empirically validate its effectiveness with different settings. Concretely, the performance comparisons are organized from different angles, including different **backbones (Tab 1)**, **datasets (Tab 2)**, **depths (Tab 4)**, and **numbers of sampled frames (Fig 1.(b))**. Moreover, we also compare our method with other advanced data augmentation strategies in Tab 3. The result shows that GM outperforms all methods except RandAugment[1] which involves multiple transformations. All of these methods are effective in image recognition, however, they have less significant improvements or even performance damage on video recognition, which indicates it is challenging to increase the video recognition generalization ability. Note that GM is compatible to the existing image-level data augmentation methods and it can improve the accuracy by **3.76%, 5.16%, 2.72%** on UCF101, HMDB51, and Something-Something V1 datasets while combining with RandAugment.
> > >
> > > Further, we want to clarify that ~1% improvement on Something-Something V1 dataset is non-trivial as even some carefully designed structures can only result in 1% improvement. For example, based on our reproduced results, TDN[2] outperforms TEA[3] by around 1% on Something-Something V1, even if TDN requires more computation and much longer training time. While training TEA with our GM, it can achieve similar performance as TDN with less costs of computational resources.
> > >
> > >
> > > **Concern2: Where improvements come from.**
> > > According to our analysis of **Tab 8**, involving GM can lead to two effects: 1) stress of motion information because parts of channels are shifted to adjacent frames; 2) alteration of the video color because of the mismatched RGB order. Therefore, we compare GM with Motion Exchange (ME) and Color Jittering (CJ) in Tab 8. The result shows that both methods lead to improvements of baseline model, but the improvement that GM results in is bigger than the increase brought by ME+CJ. It means the combination of the two effects can boost the performance of each other. We think this part elaborates well why GM works.
> > >
> > > In addition, our previous description 'GM is beneficial for temporal modeling' may be confusing. We have revised the related parts in the submitted revision into **'GM can prevent the model from overfitting non-informative frames'** which is more precise and aligned with our motivation. To further validate this point, we compare the video-wise accuracy and frame-wise accuracy (average accuracy of all sampled frames) in this table:
> > > | Method  | Top-1 Acc.(Video) | Top-1 Acc.(Frame) |
> > > | :-----: | ------------ | :-----: |
> > > | TSM | 45.34% | 27.01% |
> > > | TSM+GM | 46.48%(+1.14) | 28.29%(+1.28) |
> > >
> > > We find that GM leads to better frame-wise accuracy which means that there are more frames being correctly classified in each video sample when trained with GM and the increase of video accuracy may be attributed to this improvement. We thank the Reviewer for pointing this out during the rebuttal and we have already revised the descriptions in the submitted revision.
> > >
> > >
> > > **Concern3: Entirely concentrate on GM.**
> > > As we have mentioned in our response to Reviewer Xnqd, the motivations and implementations of Temperature-check[4] and logits smoothing are different. They regard the temperature as a tunable hyperparameter and carefully tune temperature along with learning rate and batch size to improve the performance, where they find different optimal temperatures for various architectures. Differently, in our work, we are motivated by the phenomenon that background frames may easily overwhelm the final prediction with an imbalanced distribution of logits across different frames and we simply apply the same temperature for different models to reduce the gap between logits of different frames to alleviate overfitting.
> > >
> > > We do agree with the Reviewer that logits smoothing is not our main contribution and we have already de-emphasized this part in the submitted revision. However, we still lean to keep this part even in limited space as this technique has special meaning for the overfitting problem in video recognition.

---

> > > ### Author Response · Authors · 2022-11-23
> > > **Thank you! Further response(Part 2)**
> > >
> > > **Comment1: Details of training time.**
> > > We test the training time on 2 NVIDIA Tesla V100 GPUs and we train all methods for 50 epochs for fair comparisons. We will add this part in our final version.
> > >
> > >
> > > **Comment2: Details of model name.**
> > > SlowOnly is 8 $\times$ 8 and we use Uniformer-S because of constraints in training resources.
> > >
> > > Moreover, we have conducted experiments over Uniformer[5] with 16 frames and 3-crop evaluation on Something-Something V1 and Mini-Kinetics datasets:
> > > | Method  | Top-1 Acc.(Sth-Sth V1) | Top-1 Acc.(Mini-Kinetics) |
> > > | :-----: | ------------ | ------------ |
> > > | Uniformer-S | 54.45% | 79.39% |
> > > | Uniformer-S+GM | 55.29%(+0.84) | 80.40%(+1.01) |
> > >
> > > Results show that Uniformer achieves much higher accuracy compared to the numbers in our paper and GM consistently improves the generalization ability of Uniformer with stronger performance. We will update this part in our final version.
> > >
> > > ***
> > >
> > > We appreciate the Reviewer's efforts so far for us to improve the work. We expect our further explanation can eliminate the Reviewer's concerns and we would like to make further discussions with the Reviewer about our work.
> > >
> > >
> > > Sincerely,
> > > Authors
> > >
> > >
> > > ***
> > >
> > > [1] Cubuk E D, Zoph B, Shlens J, et al. Randaugment: Practical automated data augmentation with a reduced search space[C]//Proceedings of the IEEE/CVF conference on computer vision and pattern recognition workshops. 2020: 702-703.
> > > [2] Wang L, Tong Z, Ji B, et al. Tdn: Temporal difference networks for efficient action recognition[C]//Proceedings of the IEEE/CVF Conference on Computer Vision and Pattern Recognition. 2021: 1895-1904.
> > > [3] Li Y, Ji B, Shi X, et al. Tea: Temporal excitation and aggregation for action recognition[C]//Proceedings of the IEEE/CVF conference on computer vision and pattern recognition. 2020: 909-918.
> > > [4] Agarwala A, Pennington J, Dauphin Y, et al. Temperature check: theory and practice for training models with softmax-cross-entropy losses[J]. arXiv preprint arXiv:2010.07344, 2020.
> > > [5] Li K, Wang Y, Zhang J, et al. Uniformer: Unifying convolution and self-attention for visual recognition[J]. arXiv preprint arXiv:2201.09450, 2022.

---

> > > ### Author Response · Authors · 2022-12-09
> > > **Gentel reminder for the deadline**
> > >
> > > Dear Reviewer 6srd,
> > >
> > > Thanks for your efforts so far for our paper review and we appreciate your increasing score which encourages us a lot to further improve our work.
> > >
> > > Currently, most of the concerns from Reviewer Xnqd and Xyj5 have been well-addressed and we are eager to know whether our further response has resolved your remaining concerns. Concretely, according to the newest comments, we have provided detailed explanations regarding Concern1, and we think we have already carefully revised the content regarding Concern2 and Concern3 before Nov 18 and uploaded the revised draft in the openreview system. Moreover, we have also provided much stronger results of Uniformer-S on different datasets and GM consistently improves the generalization ability of Uniformer-S which further demonstrates the effectiveness of our design.
> > >
> > > Due to the coming discussion deadline (12/12), we would like to kindly remind Reviewer 6srd if our further response has helped the Reviewer to reevaluate this work. We really appreciate it if you could give us any feedback and your opinions are rather important for us to improve the work.
> > >
> > > Thank you very much for your time!
> > >
> > > Sincerely,
> > > Authors

---

> > > > ### Comment · Reviewer_6srd · 2022-12-11
> > > > **Thank you**
> > > >
> > > > I have read and appreciate the further clarifications and comments from the authors.
> > > >
> > > > - Significance of improvements: I understand that Sth-Sth is a difficult dataset, but still expect that a model which focuses on temporal data augmentation to show a larger improvement on a temporal dataset, given the additional training costs. This is compounded by the fact that the reproduced numbers are lower than those from the original works, as pointed out by reviewer Xyj5 (even the updated UniFormer-S with multicrop eval underperforms the official result of 57.2% on SSv1). Note that random flipping itself is an effective augmentation strategy, and disabling it might contribute to model overfitting on the majority of Sth-Sth classes not sensitive to left/right directions.
> > > >
> > > > - Source of improvements: Reading the ablation section again, I am not very sure color jittering is the right comparison, since the proposed method does not involve changing the brightness/contrast. A more relevant strategy can be scrambling the RGB order within each frame (so no exchange of motion information). In general, I think this section can be better presented by breaking down to smaller tables, each validating one component of the augmentation strategy.
> > > >
> > > > I decide to keep my rating at borderline reject after reading all reviews and authors' responses. Again, I thank the authors for their engaging discussions throughout the rebuttal period, and encourage them to continue improving the work regardless of the acceptance decision.

---

> > > > > ### Author Response · Authors · 2022-12-11
> > > > > **Thank you! Final Reply**
> > > > >
> > > > > Dear Reviewer 6srd,
> > > > >
> > > > > We really appreciate the Reviewer's feedback and we understand the Reviewer decides to keep the score. Nevertheless, we still want to clarify several points based on your comments and prevent misunderstanding of our work.
> > > > >
> > > > > ***
> > > > >
> > > > > **Significance of improvements.**
> > > > > We understand the Reviewer expects a larger improvement on Sth-Sth dataset as significance is quite subjective and people hold different views. However, we want to stress that GM outperforms other data augmentation methods, except RandAugment, and all methods will increase the training time to different extent.
> > > > >
> > > > > Further, we emphasize that only the reproduced results of TAM/TDN/Uniformer-S on Sth-Sth are lower than the original works and the reasons are:
> > > > > (1) TAM/TDN: we do not apply random flipping on Sth-Sth which is also the common strategy of other works (even competing method Uniformer-S does not apply random flipping on Sth-Sth as there are certain classes being sensitive to the direction). TAM/TDN carefully pick the direction-sensitive classes and apply random flipping on the rest categories to improve the performance. We do think random flipping is effective and note that we have incorporated it with GM on the rest datasets in our paper. We will apply random flipping on carefully selected classes when training with TAM/TDN in the final version.
> > > > > (2) Uniformer-S: the possible reason may be the difference in the experimental environment and the number of GPUs as we conducted experiments on the official code and did not change any hyperparameters or training settings. Note that Uniformer-S is trained with multiple data augmentation methods, including Mixup, RandAugment, etc., and GM can still effectively improve its generalization ability, which demonstrates that the gain in performance comes from the temporal perspective as existing methods only apply spatial transformations.
> > > > >
> > > > > **Source of improvements.**
> > > > > We further conduct experiments of scrambling RGB order (SC) and the results are shown in this table:
> > > > > | Method | mAP |
> > > > > | :-----: | :-----: |
> > > > > | TSM | 73.10% |
> > > > > | TSM+ME | 73.79%(+0.69) |
> > > > > | TSM+CJ | 73.89%(+0.79) |
> > > > > | TSM+SC | 73.96%(+0.86) |
> > > > > | TSM+GM | 74.72%(+1.62) |
> > > > >
> > > > > One can observe that SC obtains similar performance with CJ. GM can be broken down to SC+ME, and the improvement that GM results in is bigger than the increase brought by SC+ME which means the combination of the two effects can boost the performance of each other. We will revise this section in the final version based on your suggestion.
> > > > >
> > > > >
> > > > > ***
> > > > >
> > > > > We appreciate the Reviewer's suggestions and for participating in the discussion till the last minute. As we said, we understand the Reviewer keeps the score and we value your feedback for us to improve the work. Thanks again.
> > > > >
> > > > > Sincerely,
> > > > > Authors

---

### Official Review · Reviewer_Xnqd · 2022-10-31

**Confidence:** 4
**Clarity, Quality, Novelty And Reproducibility:** 1) The paper is riddled with grammati…
**Correctness:** 2
**Technical Novelty And Significance:** 3
**Empirical Novelty And Significance:** 2
**Recommendation:** 6

**Strength And Weaknesses:**

Strengths:
1) The proposed data augmentation technique is well explored and has an extensive empirical evaluation. I appreciate that the authors investigated it under a number of different models and datasets. While they briefly also investigate modern approaches based on Transformers, most of their results focus on out-dated models based on 2D CNNs. I feel that the paper could be improved by focusing more on modern models. E.g. Table 1 contains results for TSN, which is by now a 6 year old method. I don't think these results have any relevance for the research community. Adding a modern, state-of-the-art method like Video Swin Transformer[4], MViT[5] or MTV[6] instead would considerably strengthen the paper.


Weaknesses
1)  The "logit scaling" part of the current manuscript is weak: it's a completely unrelated regularization technique to Ghost Motion, and as the authors do mention, the same technique was furthermore already proposed and investigated for classification tasks in [3]. So mentioning this as a novel contribution is a stretch.  Furthermore it seems to be just another variation of label smoothing [1] or logit squeezing [2], neither of which the authors compare to or even mention. I'd suggest de-empathising this part of the paper and focusing completely on Ghost Motion instead. Otherwise, the authors need to properly compare logit scaling to competing methods.

2)  The results for the baselines stated by the authors do not match those from the original publications. For example, the authors claim in Table 1 that the basic TSM module achieves 45.34% accuracy on SomethingSomethingV1, and that TSM+GM together achieve 46.88%.
However, the original TSM publication states that TSM alone achieves 47.3% accuracy. Similarly, the original TDN publication claims 52.3% top1 accuracy on SomethingSomethingV1, while this manuscript claims that TDN alone achieves 49.69% and TDN+GN  50.51%. I did not check the rest of the numbers, but this small spot check seems weird.  The authors need to address this discrepancy.

3) While the authors claim several times that GM "brings almost zero costs", that claim was never backed  up. I'd appreciate if the authors could provide insights on the computational cos of their method: does adding GM really not increase the wall-clock time of training?

4) The authors give all accuracies with 4 significant digits, which would suggest a large number of repeat experiments, yet they do neither mention this nor give error bars. I would strongly suggest they add unambiguous error bars for their results.


[1] Mueller et al., "When does label smoothing help?", NeurIPS 2019 (and references therein)

[2] Shafahi et al, "Label Smoothing and Logit Squeezing: A Replacement for Adversarial Training? ", ICLR 2019

[3] Agarwala et al, Temperature check: theory and practice for training models with softmax-cross-entropy losses, arxiv 2019

[4] Liu et al, "Video Swin Transformer", CVPR 2022

[5] Fan et al, "Multiscale Vision Transformers", ICCV 2021

[6] Yan et al, "Multiview Transformers for Video Recognition", CVPR 2022

**Summary Of The Paper:**

The authors propose a new data augmentation for video classification that consists of shifting one of the RGB channels forwards or backwards by 1 frame. They call this augmentation "Ghost Motion" (GM). They also show that scaling the logits by a temperature helps generalization. The authors claim that GM improves results for a variety of different models and datasets.

**Summary Of The Review:**

The authors propose a  input augmentation that appears to be effective on a wide range of problems and models. The empirical evaluation covers a wide range of models and datasets, yet the numbers raise some questions (see above).

UPDATE: after the author's revision of the paper, I have updated my score to reflect that I think the method now meets the standards for publication.

---

> ### Author Response · Authors · 2022-11-15
> **Response to Reviewer Xnqd(Part 1)**
>
> We thank for the Reviewer pointing out the drawback of our draft and we make the response as below.
>
> **Improvement: Results on modern methods.**
> Thanks for the comment. We conduct most of the experiments on 2D networks as our work is motivated by the analysis on them. However, we have also included the results on 3D and Transformer network Uniformer[1], which has shown stronger performance on the leaderboard of Something-Something datasets compared with Video Swin[2], MViT[3] and MTV[4]. The results in our paper are lower than the reported results in their paper as we unify the training and test recipes which sample 8 frames for all methods and adopt one-crop one-clip per video during evaluation for efficiency, considering the large amounts of experiments in our paper. Note that we have demonstrated GM can consistently improve the performance of baseline method with different depths (see Tab 4) and sampled frames (see Fig 1.(b)) which proves the generalization ability of GM on models with different representation abilities. Further, we conduct experiments over Uniformer with more sampled frames and adopt multi-crop evaluation on Something-Something V1 dataset:
> | Method  | Top-1 Acc. |
> | :-----: | ------------ |
> | Uniformer | 54.45% |
> | Uniformer+GM | 55.29%(+0.84) |
>
> One can observe that Uniformer obtains much stronger performance compared with the result in paper (41.74%) and GM still improves the accuracy by 0.84% which validates its efficacy. We admit that this is not the SOTA result in video recognition, but we have tried our best to provide results with the highest accuracy based on our computation resources.
>
>
> **Weakness 1: Logits smoothing is weak.**
> We thank the Reviewer for the valuable suggestion. First, we want to clarify that there are differences between logits smoothing and Temperature check[16]:
> (1) Motivation: They regarded temperature as a tunable temperature in Cross-Entropy loss to improve the performance of image classification models and did not consider the problem of overfitting. However, our aim is to prevent
> the background frames overwhelming the final prediction and relieve the overfitting problem in video recognition.
> (2) Implementation: In Temperature check[16], they carefully tune temperature along with learning rate and batch size to improve the performance and find different optimal temperatures for various architectures. While in our work, we simply apply the same temperature for different models ($T=3$ on Sth-Sth V1, $T=2$ on Sth-Sth V2) without further tuning the hyperparameter. We also adopt the temperature in testing as it helps to reduce the Expected Calibration Error (ECE) for calibration.
>
> However, we agree with the Reviewer that logits smoothing is not our main contribution and we regard it as an add-on technique which is orthogonal to GM to further reduce overfitting. We will reorganize the paper and try to de-emphasize this part in our final version. Thanks for the advice.
>
>
> **Weakness 2: Results do not match from original publication.**
> Thanks for asking this question. As we have explained in Sec 4.1 (Implementation details), we uniformly sample 8 frames for all methods and adopt one-crop (224 $\times$ 224) one-clip per video during evaluation for efficiency which is a common practice for 2D networks[5],[6],[7]. Besides, we did not perform random flipping on Something-Something V1&V2 datasets as there are certain classes being sensitive to the direction[5],[7]. As for the discrepancy with TSM[5], the accuracy of 47.3% is cited from Tab 1 where they use 2 clips with a resolution of 256 $\times$ 256 per video for evaluation. When they apply the same evaluation protocol with ours, the accuracy is 45.6% (Tab 2 in their paper) which is similar with our reproduced result. As for the performance gap with TANet[8] and TDN[9], the reason is that they carefully picked the direction-sensitive classes in Something-Something V1&V2 and apply random flipping on the rest categories to improve the performance. Note that we adopt the same training hyperparameters with the official codes for all methods and we believe the unified setting will not impair the fairness of comparisons.

---

> > ### Comment · Reviewer_Xnqd · 2022-11-15
> > **Reply to Rebuttal**
> >
> > I'd like to thank the authors for taking the time to write such a thorough reply, I feel like my major critiques have been well addressed. Please let me know once you have uploaded the updated PDF to openreview, and I will adjust my review score accordingly!
> >
> > I do have some remarks on your reply:
> >
> > 1) *Results on modern methods*: I'm sorry for not noticing that Uniformer actually was a strong and modern model, thank you for pointing it out and sorry for not noticing it before.
> >
> > 2) Logit Smoothing: "Reducing overfitting" is literally how the very first paragraph of your reference [16] ("Temperature Check") motivates the use of softmax temperatures, logit scaling, and similar techniques that manipulate the distribution of logits to increase generalization error (such as logit smoothing). I think focusing on GM instead would be a better choice and gives a clearer paper. Otherwise, I'd appreciate an ablation of Logit Smoothing vs the more common Label Smoothing; I don't mean to be moving goalposts/burdening you with further experiments, I will update my review score regardless of whether this is added or not. But in order to judge how well the method works, I'd personally like to know how it compares to more established and similar methods.
> >
> > 3) Results do not match from original publication: thanks for clarifying this.
> >
> > 4) Training cost of GM: "Based on these analyses, we can conclude that GM will bring negligible computational costs during training, but will result in longer training time because of the interpolation operation. " ==> I don't understand this reasoning. From this table, it seems like GM is the 2nd most expensive training method in terms of total training time, increasing the training time by 36%. So the computational cost does not seem to be negligible, at all? Or am I misunderstanding something? In any case, please include this table in the appendix or even the main paper. Knowing the cost to total training time is a crucial information for practitioners who want to apply this method.

---

> > > ### Author Response · Authors · 2022-11-17
> > > **Further response to Reviewer Xnqd**
> > >
> > > Dear Reviewer Xnqd,
> > >
> > > Thank you so much for your further advice and your reply is a huge encouragement for us.
> > >
> > > Here are our further responses to your reply:
> > >
> > > **Remark2.**
> > > We have included the experiment in the updated draft where the results show that logits smoothing and label smoothing both lead to improvement in accuracy and the performance can be further improved when combining these two methods. This may indicate that the two approaches are complementary to each other as one is smoothing the logits and the other is smoothing the label. Even though, we do agree with you that logits smoothing is not our main contribution and we have de-emphasized this part in our paper to make it clearer. Thanks so much for this suggestion.
> > >
> > > **Remark4.**
> > > Sorry for the confusion and we further clarify this point. We refer to 'computational costs' as the metric FLOPs which measures the theoretical computation. Based on our previous response, the computation of GM is 3.6 $\times$ $10^{6}$ and we regard it as negligible compared to the computation of TSM which is 32.7 $\times$ $10^{9}$. However, the training time is not merely determined by the computation and we believe it has more relation to the GPU speed of these operations. For example, depth-wise convolution has much smaller computation compared to traditional convolution, but the GPU speed of it may be even slower than traditional convolution as traditional convolution has been optimized very well on GPU devices and it can run in parallel with a faster speed. Therefore, we think the reason why GM costs more training time is that the interpolation and shifting operation have slower speed on GPU, but not caused by the computation of GM. We thank the Reviewer for pointing this out as the training time of data augmentation methods is a seldom touched topic in previous papers and we have included the discussion in our paper to provide insights for later works.
> > >
> > > As we have promised, we have revised the draft based on the suggestions from all Reviewers, including the following changes :
> > > ***
> > > * de-emphasized logtis smoothing and compared it with label smoothing[1];
> > > * added the discussion of training cost of existing data augmentation methods;
> > > * included stronger results of Uniformer[2] on Something-Something V1 dataset;
> > > * added ablation of Motion Exchange (ME);
> > > * revised the citing format;
> > > * revised the writing of certain descriptions;
> > > * corrected the typos.
> > >
> > > [1] Szegedy C, Vanhoucke V, Ioffe S, et al. Rethinking the inception architecture for computer vision[C]//Proceedings of the IEEE conference on computer vision and pattern recognition. 2016: 2818-2826.
> > > [2] Li K, Wang Y, Zhang J, et al. Uniformer: Unifying convolution and self-attention for visual recognition[J]. arXiv preprint arXiv:2201.09450, 2022.
> > > ***
> > >
> > > We have tried our best to upload this draft as early as we can and please let us know if you have further comments so that we still have a chance to improve our paper before the deadline. Thank you for being with us so far.
> > >
> > >
> > > Sincerely,
> > > Authors

---

> ### Author Response · Authors · 2022-11-15
> **Response to Reviewer Xnqd(Part 2)**
>
> **Weakness 3: Training cost of GM.**
> Good question. Basically, GM is made of two operations: shifting and interpolation. As for the computational cost, shifting operation will bring zero computation and interpolation is composed of two multiplications and one addition which costs negligible computation compared to the computation of processing the single video by the deep network (e.g., if we sample 8 frames to TSM with a resolution of 224 $\times$ 224, the costs will be 3 $\times$ 3 $\times$ 8 $\times$ 224 $\times$ 224 compared to 32.7 $\times$ $10^{9}$.)
>
> As for the wall-clock training time, we measure the GPU hours of training GM on Something-Something V1 and compare it with other image-level data augmentation methods, including Cutout[11], CutMix[12], Mixup[13], AugMix[14] and RandAugment[15] in this table:
>
> | Method | Top-1 Acc. | GPU hours |
> | :-----: | :-----: | :-----: |
> | TSM | 45.34% | 11 |
> | TSM+Cutout | 44.68% | 12 |
> | TSM+CutMix | 44.99% | 12 |
> | TSM+Mixup | 46.03% | 15 |
> | TSM+AugMix | 46.23% | 13 |
> | TSM+RandAugment | 47.20% | 12 |
> | TSM+GM | 46.88% | 14 |
>
> Though GM can effectively improve the generalization ability of baseline method, it actually results in more training hours. We further analyze the results for more insights. First, Mixup costs the most time which means the interpolation of raw data will increase wall-clock time despite their theoretical efficiency. Moreover, it costs more training time than GM which indicates that the manipulation of labels needs separate loss functions and will bring nonnegligible time costs, while GM does not touch the original label. The training time of Cutout and CutMix are less as they only process a small crop of the raw data compared to Mixup and GM, which need to do interpolation on the whole data. Though AugMix and RandAugment involve multiple augmentation operations, these transformations are traditional data augmentation techniques (e.g., rotate/contrast/translate) which can be efficiently implemented with small time increases.
>
> Based on these analyses, we can conclude that GM will bring negligible computational costs during training, but will result in longer training time because of the interpolation operation. However, as shown in Tab 6, the interpolation is indeed an essential component for GM as it offers a continuous distribution of the input data which enlarges the input space and is beneficial for recognition. We hope these analyses can provide insights for later works.
>
>
> **Weakness 4: Error bars for the results.**
> Thanks for noticing it. In our papers, we repeat the experiments in Sec 4.5 as they are used for ablation and analysis which needs more precise numbers to verify the effectiveness of specific designs. According to Tab 6, the variance is not very large so we did not repeat the experiments for results on other datasets as most of them are large-scale where the variance is relatively small. To validate this, we further repeat the experiments over TSM on Something-Something V1 for three times, and report the average accuracy and variance in this table:
>
> | Method | Top-1 Acc.(%) |
> | :-----: | :-----: |
> | TSM | 45.38 $\pm$ 0.16 |
> | TSM+GM | 46.79 $\pm$ 0.10 |
>
> We can see that the results are similar with the numbers in our paper and the variance is small. As we have validated GM on various models and datasets which need a large amount of experiments (20 experiments in Tab 1, 16 experiments in Tab 2, and 30 experiments in Tab 3), conducting multiple runs for all these experiments is quite challenging during the rebuttal phase which has limited time. However, if the Reviewer still has concerns on it, we are willing to repeat all the experiments in our paper and we are looking forward to your response.
>
>
> **Clarity.**
> We thank the Reviewer for carefully pointing out the errors in our writing. We will extensively revise the writing and upload it before Nov 18 for confirmation.

---

> ### Author Response · Authors · 2022-11-15
> **Response to Reviewer Xnqd(Part 3)**
>
> **Originality.**
> Thanks for pointing it out. As for GM, we have discussed the difference with TSM[5] in Related Work as:
> (1) Motivation: TSM is considered as a temporal modeling module which is densely inserted into the network and hard to generalize to other architectures for further improvement. GM is designed to prevent overfitting which manipulates the input data and can be generalized to different methods.
> (2) Specific Design: TSM only shifts a small portion of channels and the rest channels are kept in the original location. While GM will first reshape Channel ($C$) and Temporal ($T$) dimension into one dimension ($C \times T$), then shift all channels by one step to cause the mismatch in channel orders which brings motion exchange and color variance to alleviate overfitting.
>
>
> Then, we give the results of TSM with different temperatures on Something-Something V1 dataset:
> | Method | Top-1 Acc. |
> | :-----: | :-----: |
> | TSM(T=1) | 45.34% |
> | TSM(T=2) | 45.77%(+0.43) |
> | TSM(T=3) | 45.85%(+0.51) |
>
> We can observe logits smoothing slightly increases the validation accuracy of TSM, but the improvements are actually not significant. We further conduct experiments over TSM[5]/TANet[8]/TEA[10]/TDN[9] without logits smoothing on Something-Something V1 dataset:
> | Method  | Top-1 Acc. | Top-1 Acc.(GM) |  Top-1 Acc.(GM+smoothing) |
> | :-----: | ------------ | :-----: | :-----: |
> | TSM | 45.34% | 46.48%(+1.14) | 46.88%(+1.54) |
> | TANet | 45.94% | 46.93%(+0.99) | 47.04%(+1.10) |
> | TEA | 48.67% | 49.51%(+0.84) | 49.61%(+0.94) |
> | TDN | 49.69% | 50.32%(+0.63) | 50.51%(+0.82) |
>
> It can be seen that GM+smoothing obtains the highest accuracy among the three columns, though GM is still effective as it contributes most of the improvements over baseline methods.
>
>
>
> Again, we thank the Reviewer for the constructive suggestions which help us to improve the work. We are actively available until the end of this rebuttal period and let us know if you have any further questions. Looking forward to hearing back from you.
>
>
> [1] Li K, Wang Y, Zhang J, et al. Uniformer: Unifying convolution and self-attention for visual recognition[J]. arXiv preprint arXiv:2201.09450, 2022.
> [2] Liu Z, Ning J, Cao Y, et al. Video swin transformer[C]//Proceedings of the IEEE/CVF Conference on Computer Vision and Pattern Recognition. 2022: 3202-3211.
> [3] Fan H, Xiong B, Mangalam K, et al. Multiscale vision transformers[C]//Proceedings of the IEEE/CVF International Conference on Computer Vision. 2021: 6824-6835.
> [4] Yan S, Xiong X, Arnab A, et al. Multiview transformers for video recognition[C]//Proceedings of the IEEE/CVF Conference on Computer Vision and Pattern Recognition. 2022: 3333-3343.
> [5] Lin J, Gan C, Han S. Tsm: Temporal shift module for efficient video understanding[C]//Proceedings of the IEEE/CVF International Conference on Computer Vision. 2019: 7083-7093.
> [6] Wang L, Xiong Y, Wang Z, et al. Temporal segment networks: Towards good practices for deep action recognition[C]//European conference on computer vision. Springer, Cham, 2016: 20-36.
> [7] Wang Y, Chen Z, Jiang H, et al. Adaptive focus for efficient video recognition[C]//Proceedings of the IEEE/CVF International Conference on Computer Vision. 2021: 16249-16258.
> [8] Liu Z, Wang L, Wu W, et al. Tam: Temporal adaptive module for video recognition[C]//Proceedings of the IEEE/CVF International Conference on Computer Vision. 2021: 13708-13718.
> [9] Wang L, Tong Z, Ji B, et al. Tdn: Temporal difference networks for efficient action recognition[C]//Proceedings of the IEEE/CVF Conference on Computer Vision and Pattern Recognition. 2021: 1895-1904.
> [10] Li Y, Ji B, Shi X, et al. Tea: Temporal excitation and aggregation for action recognition[C]//Proceedings of the IEEE/CVF conference on computer vision and pattern recognition. 2020: 909-918.
> [11] DeVries T, Taylor G W. Improved regularization of convolutional neural networks with cutout[J]. arXiv preprint arXiv:1708.04552, 2017.
> [12] Yun S, Han D, Oh S J, et al. Cutmix: Regularization strategy to train strong classifiers with localizable features[C]//Proceedings of the IEEE/CVF international conference on computer vision. 2019: 6023-6032.
> [13] Zhang H, Cisse M, Dauphin Y N, et al. mixup: Beyond empirical risk minimization[J]. arXiv preprint arXiv:1710.09412, 2017.
> [14] Hendrycks D, Mu N, Cubuk E D, et al. Augmix: A simple data processing method to improve robustness and uncertainty[J]. arXiv preprint arXiv:1912.02781, 2019.
> [15] Cubuk E D, Zoph B, Shlens J, et al. Randaugment: Practical automated data augmentation with a reduced search space[C]//Proceedings of the IEEE/CVF conference on computer vision and pattern recognition workshops. 2020: 702-703.
> [16] Agarwala A, Pennington J, Dauphin Y, et al. Temperature check: theory and practice for training models with softmax-cross-entropy losses[J]. arXiv preprint arXiv:2010.07344, 2020.

---

### Decision · Program_Chairs · 2023-01-20

**Decision:**

Reject

**Justification For Why Not Higher Score:**

(1) Most of the architectures considered were not modern so it's unclear whether the benefits of this method transfer to the practically relevant setting.
(2) Logit scaling is essentially orthogonal and already well-understood aspect of this work, and puts many results outlined in the paper under question as it's unclear how to disentangle the performance gain.
(3) Motivation behind Ghost Motion in terms of why would this specific design choice bring significant practical benefits.

**Justification For Why Not Lower Score:**

N/A

**Metareview: Summary, Strengths And Weaknesses:**

The authors propose a new method for data augmentation, Ghost Motion, applicable to video understanding problems, such as action recognition. The proposed method synthesizes the augmented video by taking a video and applying a random shift in time by 1 frame to one of the RGB channels. The authors also investigate temperature scaling as a technique to reduce overfitting in this setting. The empirical evaluation shows that the proposed method improves performance across multiple architectures and downstream tasks.

This was a borderline submission. The reviewers appreciated the simplicity of the proposed method, somewhat extensive evaluation, and the fact that this problem is often overlooked. They also pointed out several weaknesses: (1) Most of the architectures considered were not modern so it's unclear whether the benefits of this method transfer to the practically relevant setting. (2) Logit scaling is essentially orthogonal and already well-understood aspect of this work, and puts many results outlined in the paper under question as it's unclear how to disentangle the performance gain. (3) Motivation behind Ghost Motion in terms of why would this specific design choice bring significant practical benefits.

During the rebuttal some of these issues were partially addressed and some reviewers improved their score. Nevertheless, at the end of the discussion the general consensus remained -- the submission is still borderline and the reviewers feel that the work would mostly benefit a more focused audience. I will hence recommend rejection and suggest the authors to improve on (1-3), ideally removing the confounding factor (2), and provide a more detailed analysis in terms of running time impact across all the benchmarks.